# WaveSense: Efficient Temporal Convolutions with Spiking Neural Networks for Keyword Spotting

## Abstract

Ultra-low power local signal processing is a crucial aspect for edge applications on always-on devices. Neuromorphic processors emulating spiking neural networks show great computational power while fulfilling the limited power budget as needed in this domain. In this work we propose spiking neural dynamics as a natural alternative to dilated temporal convolutions. We extend this idea to WaveSense, a spiking neural network inspired by the WaveNet architecture. WaveSense uses simple neural dynamics, fixed time-constants and a simple feed-forward architecture and hence is particularly well suited for a neuromorphic implementation. We test the capabilities of this model on several datasets for keyword-spotting. The results show that the proposed network beats the state of the art of other spiking neural networks and reaches near state-of-the-art performance of artificial neural networks such as CNNs and LSTMs.

## 1 Introduction

Local signal processing is an important component of the computational pipeline for Internet-of-Things (IoT) devices equipped with a range of sensors like audio, video, and motion sensing. A significant range of these sensors capture signals comprising temporal features. Ideally these features need to be extracted by an on-board processor before decision making or relaying the pre-processed information for further computation. Processing temporal signals is often computationally challenging and requires large amounts of memory and power, especially in always-on scenarios. Neuromorphic (Mead, 1990) processors with spiking-neural networks have shown promise in this domain as ultra-low power compact solutions Indiveri et al. (2011); Benjamin et al. (2014); Merolla et al. (2014); Furber et al. (2014); Davies et al. (2018); Liu et al. (2019).

In this work we propose an elegant way of implementing temporal convolutions in spiking neural networks by leveraging their inherent synaptic and membrane dynamics. Based on this idea we propose WaveSense, a Spiking Neural Network (SNN) model suitable for efficient neuromorphic implementations while retaining high accuracy on temporal data streams. This work bridges the performance gap between Artificial Neural Networks (ANNs) and SNNs for temporal tasks. Crucially, the proposed model

- accepts spike streams and not 'buffered frames' as input,
- requires no delays in its connectivity,
- utilizes a very simple spiking neuron model - Leaky Integrate and Fire (LIF) neuron - without the need of any additional adaptive mechanisms,
- does not require recurrent connectivity (which can often be difficult to tune or train),
- achieves a high classification performance.

Recently several works have shown how to build efficient SNNs with accuracies equivalent to ANNs Diehl et al. (2015); Rueckauer et al. (2017). In these studies, spiking neurons are used in rate mode with equivalent response curves to ReLU activations, to transfer weights from pre-trained ANNs to SNNs. This approach therefore completely neglects the temporal capabilities of spiking neurons. On the other hand, surrogate gradient methods enable directly training SNN using Back Propagation

Through Time (BPTT) Neftci et al. (2019). (Shrestha & Orchard, 2018) for instance show temporal processing on temporal gesture recognition task and show a good performance on a visual task. Similar approaches have already been investigated on audio tasks Bellec et al. (2018); Wu et al. (2019); Cramer et al. (2020). (Bellec et al., 2018) demonstrate classification results on the TIMIT dataset using long time constants, a complex learning strategy (Deep-R) and require significant computational resources to train. (Wu et al., 2019) train SNNs for automatic speech recognition tasks in a tandem approach with an ANNs. This training pipeline integrated a language model and pronunciation model which goes beyond the capabilities of the neuromorphic system. The same authors showed in an earlier study the capabilities of an SNN in combination with a self-organized map to learn to recognize digits using the TIDIGIT dataset Wu et al. (2018).

(Blouw et al., 2018) demonstrate high accuracy in a audio classification task using dense networks with spectrograms as inputs. This approach requires passing the frequency data of previous time steps (defined by the spectrogram time window) in every sample presentation to the network. (Kugele et al., 2020) show that by matching ANN roll-out delays to the propagation delays in SNN, the resultant networks can demonstrate a high accuracy on vision-based spatio-temporal classification tasks. Implementation of delays in neuromorphic hardware requires additional memory resources to store and deliver spikes in a delayed fashion and could be potentially quite expensive. (Yin et al., 2020) use a Spiking Recurrent Neural Network (SRNN) architecture and demonstrate that by utilizing *adaptive* LIF neurons and learning the time constants, these networks can perform temporal classification tasks in a sequential manner fairly well. The authors demonstrate the effective use of spiking neural networks and show a significant increase in power efficiency. Unfortunately, having fine tuned time constants in low-power neuromorphic hardware can often be challenging especially while using fixed precision numerical representations and computations.

We propose a novel network architecture for SNN that does not require buffering or delays and can directly process temporally varying streams of spiking data from event-based sensors using simple LIF neurons. Our architecture is derived from first principles and inspired by the WaveNet van den Oord et al. (2016) architecture that does not necessitate learning the time constants of the system but could be defined as the task demands. In addition we also propose an efficient training strategy and a corresponding loss-function that is suitable for streaming based models, in particular models that could be run in real-time with neuromorphic hardware.

The first key aspect of the WaveNet architecture is the use of multi-layer causal dilated convolutions. The *causal* refers to the use of data from the past, *dilated* refers to a sparse kernel and the *convolution* is along the time axis. Stacking such convolutional operations along multiple layers enables the network to have a long temporal memory. A second aspect is that it eliminates the need for sliding window based inference/prediction and minimizes the number of computations within the network when operating on a continuous stream of data.

The WaveNet architecture is very amenable to general purpose micro-processors and micro-controllers but it still requires a reasonable amount of memory and non-linear computations such as $tanh$ and $sigmoid$ which are fairly complex (within the context of ultra-low power devices) and this in turn requires higher energy and power requirements. Neuromorphic technology promises to bring the energy required for these tasks down by utilizing SNNs to perform ultra-low power computation. So far, while neuromorphic devices have been demonstrated to operate at extremely low power, they have fallen short at demonstrating computational performance that is on-par or comparable to state-of-the-art ANNs for temporal tasks, most prominently in the audio domain.

The WaveSense model proposed here aims to bridge this gap. For the results in this work, we focus on audio tasks as spatio-temporal tasks without loss of generality. Sec. 2 details all the methods used for audio data pre-processing, conversion to spikes and the details of the network architecture. In Sec. 3 we demonstrate the computational capability of this network over several audio datasets for key-word spotting tasks. We compare our results to the state-of-the-art SNNs and ANNs. We conclude in Sec. 4 where we discuss the implications and impact of this work and potential areas where this work could be utilized.

Most importantly all the code used to generate the reported results have been made open-source and can be found at `http://XXXXXXXXXXXXX`. We believe this will enable the research community to explore other avenues that could take advantage of our work.

## 2 MATERIALS AND METHODS

### 2.1 DILATED TEMPORAL CONVOLUTIONS

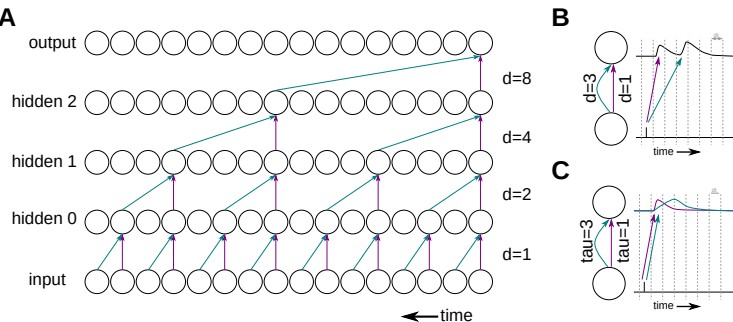

Figure 1: Relegating the job of delays in dilated convolutions to synaptic dynamics.

Temporal convolutions enable a layer of neurons to integrate information from the past. The dilation enables convolution over a large time window (or sample length) while still only using a small number of parameters. Temporal dilated convolutions therefore, perform a weighted-accumulation of information from different points in time, separated by the *dilation* parameter. This is done by storing previous activations in ANNs. Naively, the equivalent could be achieved in SNNs by utilizing synaptic transmission delays. In this work, we observe that neuron and synaptic dynamics could be seen as proxies for temporal convolutional processing as shown in Figure 1. Figure 1B shows the contributions of each projection with a kernel size 2 when implemented with delays. Implementing synaptic transmission delays in real-time neuromorphic hardware incurs a steep overhead in terms of memory and computational resources and only a limited number of neuromorphic devices support them.

We instead propose to use an appropriate set of synaptic time constants $\tau_s$ as shown in Figure 1C. While quantitatively, these are different, qualitatively both these approaches provide the ability to transform and project information in the temporal domain Sheik et al. (2012). This is the key insight we leverage to design our final model.

### 2.2 NETWORK DESCRIPTION

We take inspiration from the work of (Coucke et al., 2018) who uses the WaveNet architecture van den Oord et al. (2016) for classification of continuous audio streams. The WaveNet architecture provides a prescription for distributing temporal memory and computation across layers without repeated presentation of previous input data.

The original ANN WaveNet model comprises of a few different computational building blocks. We translate each of these building blocks to SNNs as follows.

**Dilated Temporal Convolutions** as mentioned above are implemented by synaptic projections with multiple time constants.

**Rectified Linear Unit (ReLU)** activations can be *approximated* by spiking neurons because a spiking neuron can only produce spikes if the membrane potential crosses a threshold and is silent otherwise Diehl et al. (2015).

**Non-linear activations** like $tanh$ and $sigmoid$ cannot be efficiently translated to SNNs. So we choose to replace these activations with SNNs activations (potentially ignoring the benefits of filtering and gating). In the original model, these two activations are preceded by two sets of weights, where as in our SNN we only use one weighted projection. (See Figure 2)

**Residual connections** and summation (+) are realized by a synaptic connection.

The WaveSense model is built upon these building blocks as shown in Figure 2. It comprises of several 'blocks', each of which comprises of three spiking neuron layers. The first spiking layer receives inputs filtered by two separate synapses with different time constants $\tau_s$ and weights. The

time constants $\tau_s$ of the slow projections in each of these blocks are chosen such that they span a range of values relevant to the task. The number of blocks is chosen such that the sum of all these time constants is proportional to the temporal memory demanded by the task. This layer projects to the second spiking layer. Additionally a third spiking layer in each of these blocks projects to a 'hidden' layer followed by a non-spiking 'low pass' readout layer. The output of this block is the summation of its input (residual connection) and the output of the second layer. These 'blocks' are connected in a feed forward manner. The non-spiking 'low-pass'(LP) layer simply acts as a weighted low pass filter on the spikes of the 'hidden' layer. This is equivalent to the synapse of a spiking neuron (without the neuron's membrane potential or the spiking dynamics) and does not require any extra components unavailable to spiking neurons on a typical neuromorphic platform. The choice of leaving the output layer to be non-spiking is to enable a smooth, continuous valued readout useful for faster learning using BPTT.

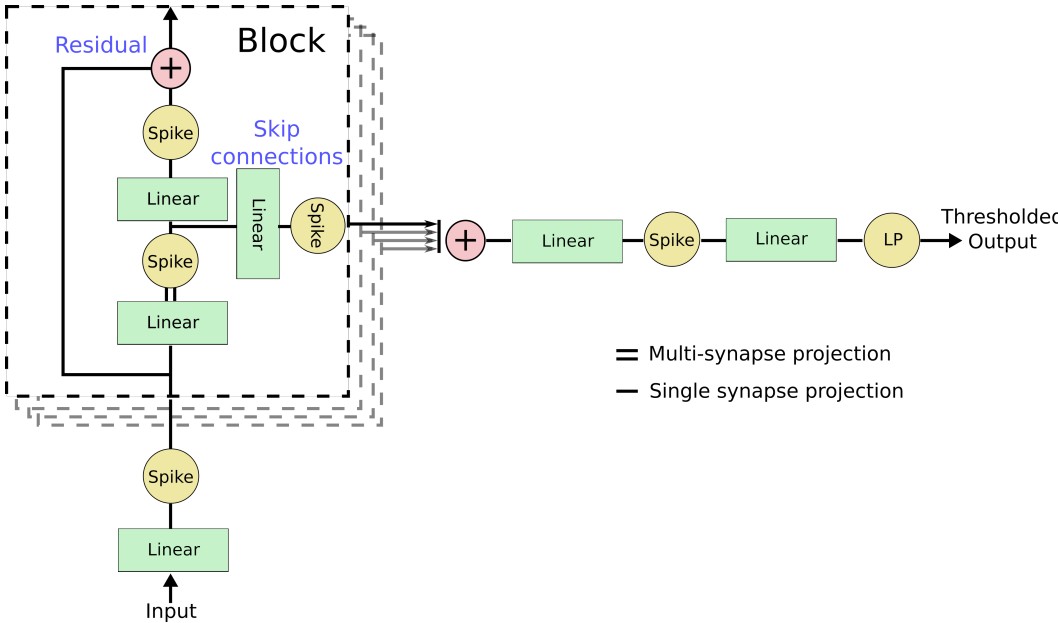

Figure 2: The WaveSense model prescribed in this work is a theoretical adaptation of the WaveNet architecture van den Oord et al. (2016) based on first principles.

## 2.3 DATASETS

In order to evaluate the efficacy of the proposed model, we train and test it against several open-source publicly available audio datasets.

### 2.3.1 ALOHA DATASET

The Aloha dataset Blouw et al. (2018) is a small collection of audio samples containing the keyword 'Aloha' and several distractors such as 'take a load off'. As the dataset is very small, only $\sim 2000$ samples, we augmented the samples using the MUSAN noise dataset Snyder et al. (2015). For that end we standardized the sample length of each utterance in the training and validation set to five seconds and added randomly selected background noise data with a signal-to-noise ratio (SNR) of 5 dB to the training data.

### 2.3.2 HEY SNIPS DATASET

The 'Hey Snips' dataset Coucke et al. (2018) for wake phrase spotting distinguishes between two classes. The positive class contains 11'000 utterances from over 2'000 speakers of the wake phrase 'Hey Snips' while the negative (or distractor) class contains over 86'000 negative examples from more than 6'000 speakers. We split the data into a training-, validation- and test-set as provided by

the authors of the dataset. We standardized the sample length of each utterance in the training and validation set to five seconds. As the dataset is already very large, no noise augmentation was needed.

### 2.3.3 SPEECH COMMANDS DATASET

The Speech Commands Warden (2018) describe a dataset containing 35 keywords uttered in total 105'000 times from over 2'600 speakers. The keywords contain the numbers 0 - 10, commands such as "stop", "go", "left" and "right" as well as other words like "Marvin", "Sheila", etc. This dataset was initially designed for keyword spotting in a limited vocabulary and the intended experiment is to detect 10 commands (plus silence) out of all 35 keywords (12 classes in total). Nevertheless, there are studies training models and showing results on all 35 classes Cramer et al. (2020). We augment the training set with noise data from the MUSAN dataset using an SNR of 5 dB just as we do for the Aloha dataset.

### 2.4 PRE-PROCESSING

The raw audio data is pre-processed in several stages:

- **Noise augmentation** The training data is augmented with noise from the MUSAN noise dataset using a SNR of $5$dB (except for the HeySnips data).
- **Length standardization and pre-amplification** The noise augmented waveform is cut into a standard length (dependent on the dataset) and the amplitude is normalized.
- **Band-pass filters** The audio is then passed through $64$ Butterworth bandpass filters of 2nd order. The bandpass filters are distributed in Mel-scale between $100$Hz and $8k$Hz.
- **Rectification** The response of the $64$ bandpass filters is rectified using a full-wave rectifier.
- **Spike conversion and binning** The output of the rectifier is applied as direct input to the membrane of $64$ simplified LIF neurons resulting in a rate code. The spike trains are binned into $10$ms timesteps allowing multiple spikes per timestep.

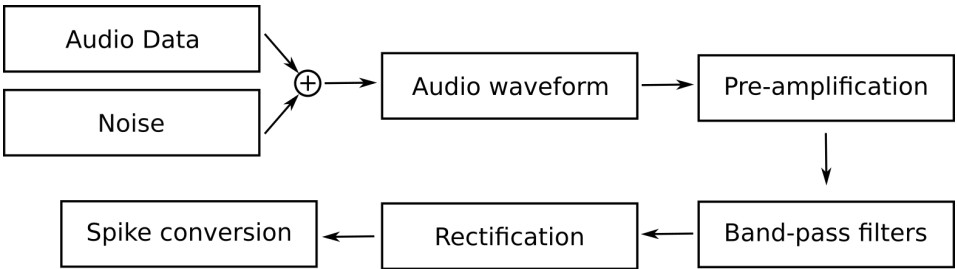

Figure 3: Data pre-processing pipeline figure.

### 2.5 TRAINING METHOD

In order to train the parameters of the SNN (See Sec. 2.2) we use BPTT. In particular we aim to be able to deploy the network in streaming mode *i.e.* the model receives the data stream directly generated from a sensor without any frame-based (sliding window) buffering. This requires us to employ an appropriate loss function.

Often in a classification task, the output class can be determined by computing cross-entropy loss on the sum of the outputs over the sample length for each output neuron. While this would yield a good classification accuracy, the magnitude of the output trace at 'a given point in time' is not indicative of the network prediction. This is not ideal for models being run in streaming mode.

### 2.5.1 PEAK LOSS

Typically for streaming models, a signal is predicted as belonging to a certain class when the corresponding output trace exceeds a 'detection threshold'. This approach is also ideal for always-on

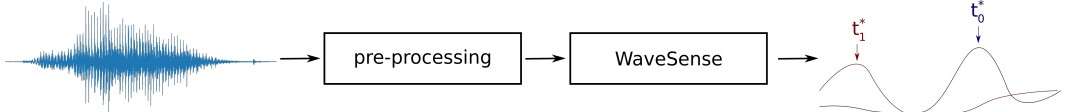

Figure 4: An example visualization of the peak loss with peak times $t_c^*$ for channels 0 and 1.

neuromorphic systems. We therefore design our loss function to reflect this detection mechanism and train our neural networks. We determine the peaks of the output traces and use only the activation values at the peaks to compute the cross entropy loss (see Figure 4) similar to *max-over-time loss* Cramer et al. (2020).

Consequently the loss is computed as follows:

$$L_{CE} = -\sum_c \lambda_c \log(p_c) \tag{1}$$

where $\lambda_c$ yields 1 if class label $c$ corresponds to the current input and 0 otherwise. $p_c$ is the prediction probability by the neural network that the current input belongs to class $c$. It is calculated by a *softmax* operation as shown below.

$$p_c = \frac{e^{\hat{\mathbf{y}}_c}}{\sum_i e^{\hat{\mathbf{y}}_i}} \tag{2}$$

where $\hat{\mathbf{y}}$ are the 'logits' produced by the neural network.

For temporal tasks, the input $\mathbf{x} = x^T = x^{1...T}$ and the output (logits) $\hat{y}$ of the neural network are time-series over time $T$.

$$\hat{y}^t = f(x^t|\Theta, s^t) \tag{3}$$

where $f$ is the transformation of the neural network, $\Theta$ are the network parameters and $s^t$ is the internal state of the network at time $t$. In *peak-loss* we pass the peak of each output trace to the *softmax* function. The peaks are calculated as follows:

$$\hat{\mathbf{y}}_c = \max(y_c^T) = y_c^{t_c^*} \tag{4}$$

where $t_c^* = \text{argmax}(y_c^T)$ is the 'peak time', the time of maximal activation of output trace $c$ (see Figure 4).

### 2.5.2 Spiking activity regularization

The activity of LIF neurons can change dramatically during the learning process. It can either lead to the absence of spikes which stalls learning or in exploding activation which results in high energy utilization of the network in a neuromorphic implementation.

In order to limit the activity of these neurons and maintain sparse activity, we include an activity regularizer term in our loss function Sorbaro et al. (2020).

$$L_{act} = (N_{spk}^\dagger/(T \cdot N_{neurons}))^2 \tag{5}$$

where the activation loss $L_{act}$ is dependent on the total *excess* number of spikes $N_{spk}^\dagger$ produced by the network with a population size $N_{neurons}$ in response to a input of length $T$ time steps. $N_{spk}^\dagger$ is given as:

Table 1: Aloha result model size and resource comparison.

| Publication | #Neurons | #Parameters | Accuracy |
|---|---|---|---|
| (Blouw et al., 2018) | 541 | 172800 | 95.8 |
| This work | 864 | 18482 | $98.0 \pm 1.1$ |

$$N_{spk}^{\dagger} = \sum \sum_i N_i^t \Theta(N_i^t - 1) \qquad (6)$$

is the sum of spikes from all neurons $N_i$ exceeding 1 in each time bin $t$ ($\Theta$ is a heaviside function).

Finally the loss function is given as:

$$L = L_{CE} + \alpha L_{act} \qquad (7)$$

where $\alpha$ was chosen to be $0.01$.

## 3 RESULTS

In order to validate and verify that sufficient information from the input is retained after pre-processing and conversion to spikes, we train a state-of-the-art WaveNet classifier on the datasets considered in this work and check that we can obtain a high accuracy. We implement a non-spiking dilated Convolutional Neural Network (CNN) to replicate the WaveNet architecture very similar to that described in (Coucke et al., 2018; van den Oord et al., 2016) (see Section 2.2 for details).

We train this ANN on the HeySnips, Aloha and SpeechCommands datasets and compare our results to those reported in literature Coucke et al. (2018); Blouw et al. (2018); Cramer et al. (2020). The results obtained from this network are then used as baseline to evaluate the performance of the proposed SNN.

### 3.1 ALOHA DATASET

In order to compare our model to other SNN implementations in the keyword spotting domain, we trained our WavseSense on the Aloha dataset Blouw et al. (2018). Table 1 shows the memory resources of the proposed model in comparison to the work demonstrated in (Blouw et al., 2018). With an average accuracy of $98.0\%$ with a standard deviation of $1.1\%$, the model presented in this work performs significantly better while at the same time requiring a significantly fewer parameters. The best runs of the WaveSense model yielded $99.5\%$ accuracy which is equal to the performance of the ANN model. It is important to note that the key focus of the work by (Blouw et al., 2018) is to benchmark energy and power consumption and not model performance.

### 3.2 HEYSNIPS DATASET

On the HeySnips dataset, our implementation of the WaveNet reaches an accuracy of $99.8\%$ on the clean dataset. In (Coucke et al., 2018), the authors do not report any accuracy number but rather report the false rejection rate (FRR) of $0.12\%$ for a fixed false alarm per hours (FAPH) of $0.5$. In order to compare our results more accurately, we implement the same metrics; our WaveNet implementation reaches $0.95$ FAPH and a $0.8\%$ FRR on the test set. These results are slightly worse than the results reported by (Coucke et al., 2018) but that is expected as we do not apply the same specific methods to improve performance such as "End-Of-Keyword labeling" and "masking". Without those methods and without gating, the FRR reported by (Coucke et al., 2018) drops to $0.98\%$. On the other hand, our WaveNet implementation reaches similar or even better results than the CNN and LSTM reported by (Coucke et al., 2018). This fact shows that our pre-processing method indeed extracts sufficient information from the input such that a neural network can reach very high accuracy. Hence, we train a spiking version of the WaveNet architecture (WaveSense), as described in 2.2, on the same data.

In the WaveSense model we do not use any gating mechanism, a kernel size of 2 and only 8 layers; much less compared to the 24 layers and kernel size of 3 as used in the WaveNet implementation by

Table 2: A comparison of model performance for various datasets and network architectures.

| Publication | Dataset | Accuracy (%) | Architecture |
|---|---|---|---|
| (Coucke et al., 2018) | HeySnips | FRR 0.12 FAPH 0.5 | WaveNet |
| (Coucke et al., 2018) | HeySnips | FRR 2.09 FAPH 0.5 | LSTM |
| (Coucke et al., 2018) | HeySnips | FRR 2.51 FAPH 0.5 | CNN |
| This work | HeySnips | 99.8 (FRR 0.8 FAPH 0.95) | WaveNet |
| This work | HeySnips | $99.6 \pm 0.1$ (FRR 1.0 FAPH 1.34) | SNN |
| (Cramer et al., 2020) | SpeechCommands(35) | $50.9 \pm 1.1$ | SNN |
| (Cramer et al., 2020) | SpeechCommands(35) | $73 \pm 0.1$ | LSTM |
| (Cramer et al., 2020) | SpeechCommands(35) | $77.7 \pm 0.2$ | CNN |
| (Perez-Nieves et al., 2021) | SpeechCommands(35) | $57.3 \pm 0.4$ | SNN |
| This work | SpeechCommands(35) | 87.6 | WaveNet |
| This work | SpeechCommands(35) | $79.6 \pm 0.1$ | SNN |
| (Blouw et al., 2018) | Aloha | 93.8 | SNN |
| This work | Aloha | 99.5 | WaveNet |
| This work | Aloha | $98.0 \pm 1.1$ | SNN |

(Coucke et al., 2018). The memory in our model is still long enough as WaveSense implements the dilations using synaptic dynamics with long time constants but the number of parameters drops from $47'090$ to $13'042$. Despite the low number of parameters and quantization from spiking activations, the WaveSense model achieves an average accuracy of $99.6\%$ over 11 runs (only drops by $0.2\%$). Our best run of the WaveSense model yielded the same accuracy (of $99.8\%$) as our WaveNet implementation. With an FRR $= 1.0\%$ and FAPH $= 1.34$ the performance is indeed lower than the WaveNet, but it is comparable to that of LSTM and CNN as reported by (Coucke et al., 2018).

### 3.3 SPEECHCOMMANDS DATASET

We also trained WaveSense on the SpeechCommands dataset. We evaluated our model by training it to classify all 35 classes in the dataset. In a study by (Perez-Nieves et al., 2021), in which the authors investigate the impact of heterogenity of time constants on the performance, the best model reached $\sim 57.3\%$ accuracy on the same dataset. In (Cramer et al., 2020) the best performing SNN is a recurrent network which yields $\sim 50.9\%$ accuracy of all 35 classes. In the same study, also an LSTM and CNN are trained on the same data resulting in an accuracy of $\sim 73\%$ resp. $\sim 77.7\%$. The WaveSense model reaches an average accuracy of $79.6\%$ over 11 runs (best $80.0\%$) which is significantly higher than the best SNN described in previous studies. Notably, WaveSense performs better than the reported LSTM and CNN Cramer et al. (2020).

## 4 DISCUSSION AND CONCLUSION

While the results demonstrated here are obtained using a fixed set of time constants, it is conceivable that according to the constraints of the neuromorphic hardware, an appropriate network could be trained to obtain qualitatively similar results. This holds true even for mixed-signal neuromorphic devices Indiveri et al. (2011) with programmable weights and tune-able time constants. Because the algorithm provides a recipe for how to choose the time constants in the network, even if a neuromorphic substrate has a limited range of time constants, a number of layers with an appropriate combination (sum) of time constants can always be chosen to fit the temporal task. This is in stark contrast to recurrent neural networks that often require a tight balance between excitation and inhibition and long time constants Bellec et al. (2018); Yin et al. (2020).

The choice of time constants and number of layers is informed by the total temporal memory required by the task. We choose them in a similar fashion to that of WaveNet with time constants increasing with factors of 2 and such that the sum of all the time constants is proportional to $\tau_{task}$. Typically we observe that a proportionality of 2.5 is suitable with a kernel size of 2. The proportionality factor is the length of time after which the effect of a Post Synaptic Potential (PSP) is negligible. This also translates to compact networks with fewer parameters for the same amount of temporal memory (at the same time resolution). In other words, given a network, the temporal memory of a given task can be computed as follows:

$$\tau_{task} \approx 2.5 \sum_i \tau_s^i \qquad (8)$$

where $i$ is the list of all the layers in the WaveSense network.

While the results reported here are significantly high, we believe this can be further improved by modifying the loss function. For instance the peak loss computed only during the presence of a keyword as opposed to the entire sample Coucke et al. (2018) has been shown to improve performance of such models. Furthermore, a thorough architecture search could potentially result in a better combination of time constants $\tau_m$ and $\tau_s$, number of channels, kernel sizes etc.

A crucial factor in adopting a model is ease of training, deployment and power efficiency. By utilizing simple LIF neurons, we take full advantage of their computational efficiency Yin et al. (2020) in additional to sparse computations afforded by SNNs. While training SNNs is relatively slow on CPUs and GPUs, utilizing the Spike Response model (SRM) in combination with the SLAYER algorithm Shrestha & Orchard (2018), we are able to train at a relatively high speed. All experimental results reported in this manuscript were performed on a single NVIDIA 1080 Ti with a few hours of run time per experiment. We further improve upon this efficiency with a custom fork of the SLAYER implementation [1]. The resulting models, while accurate within the SRM framework, are not *identical* to simulations based on LIF neurons, supported by most digital neuromorphic devices. But we find that they are a close approximation and a quick retraining can recover the model's performance using LIF neurons.

The WaveNet architecture requires storing activations of each of its layers depending on their kernel size and dilation value: $N_{buf} \propto (k-1) \cdot d + 1$. In contrast, WaveSense does not buffer any spikes(activations) from the past explicitly. Instead, the information is retained in the neuron and synaptic states: $N_{buf} \propto k + 1$. This makes WaveSense extremely efficient in terms of memory utilization in contrast to WaveNet.

The results demonstrated here show that the WaveSense architecture is suitable for audio classification tasks and show a promising performance improvement in comparison to prior state-of-the-art. Audio signals, after they are pre-processed are equivalent to a population of neurons producing spike patterns with complex spatio-temporal correlations. We argue therefore, that the results presented here can be extended to other modalities of sensory data such as ECG, PPG, machine vibrations or DVS data.

This work we believe could contribute towards a future with a ubiquitous abundance of always-on audio and other sensory devices responding to user commands. This could lead to potential misuse of the technology for surveillance. Thankfully, neuromorphic algorithms such as the one proposed here require specialized neuromorphic hardware to take full advantage. If the availability of such hardware could be regulated, we hope that society can benefit from this technology while protecting itself from misuse.

---

[1] https://XXXXXXXXXXXXX

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

# A APPENDIX

## A.1 NEURON MODEL

In this work we used the Leaky Integrate and Fire (LIF) neuron model with synaptic time constant $\tau_s$ and membrane time constant $\tau_v$. The sub-threshold dynamics of this neuron are described below.

$$\dot{v}(t) = -v(t)/\tau_v + i_s \tag{9}$$

$$\dot{i}_s(t) = -i_s(t)/\tau_s + \sum w_j s_j(t) \tag{10}$$

where $v$ is the membrane potential, $i_s$ is the synaptic current, $w$ the synaptic weight and $s$ is the input spike train.

In-order to optimize simulation time and computational efficiency, we make some alterations. Unlike a traditional LIF which resets to a resting potential upon reaching threshold $\theta$, we *subtract* a fixed value $\theta$ from the membrane potential. Furthermore, if the membrane potential increases beyond $N\theta$, where $N$ is an arbitrary positive integer, then this neuron produces $N$ spikes, and proportionally $N\theta$ is *subtracted* from the membrane potential.

$$s(t) = \begin{cases} [v(t)/\theta], & \text{if } v(t) \geq \theta \\ 0, & \text{otherwise} \end{cases} \tag{11}$$

The use of a mechanism for generating multiple spikes enables the computation to be more robust to the choice of simulation time steps. This enables us to choose a relatively large time step for our simulations. In practice we observe that some neurons occasionally *do* produce multiple spikes. (Producing multiple spikes in a single simulation time-step is not necessary if one chooses a small enough time step.)

For further simulation efficiency the LIF neurons were simulated using the SRM Gerstner (2001).

$$v(t) = \sum_j w_j (\epsilon * s_j)(t) + (\nu * s)(t) \tag{12}$$

where

$$\epsilon_s(t) = e^{(-t/\tau_s)} \tag{13}$$

$$\epsilon_v(t) = e^{(-t/\tau_v)} \tag{14}$$

$$\nu(t) = -\theta \, e^{(-t/\tau_v)} \tag{15}$$

$$\epsilon(t) = (\epsilon_s * \epsilon_v) \tag{16}$$

We use exponential kernels for synaptic $\epsilon_s(t)$ and membrane dynamics $\epsilon_v(t)$ and derive the PSP kernel $\epsilon(t)$. The refractory kernel $\nu(t)$ is also a negative exponential kernel with the same time constant $\tau_v$ as the membrane potential. The symbol $*$ denotes a convolution operation.

As spike generation is non-differentiable, we use a surrogate gradient Neftci et al. (2019). Several profiles for the surrogate gradients have been proposed in literature. We use a modified exponential kernel for the surrogate gradient function. In order to accommodate the multi-spike behavior of the neuron, we choose a periodic exponential function (Figure 5) as the surrogate gradient. This function peaks as the membrane potential approaches multiples of neuron spiking threshold $\theta$. Intuitively, this gradient function maximizes the impact of a parameter when the neuron is close to spiking or has just spiked and is a variant of the exponential gradient function. An extreme simplification of the periodic exponential would be a Heaviside function [2].

---

[2] The heaviside function like a ReLU has a range of membrane potentials where the gradient is 0 and could potentially prevent the network from learning at low activity levels.

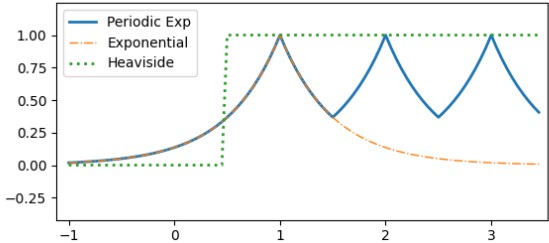

Figure 5: The surrogate gradient of the spiking neuron as a function of the (pre-spike) membrane potential.

The simulations were run with the library 'sinabs' Sheik & Sorbaro (2013)[3] using an adaptation of SLAYER Shrestha & Orchard (2018).

## A.2 SIMULATION PARAMETERS

The various parameters used in the experiments described in this article are listed below.

Table 3: Parameters used for the Aloha simulations (ANN).

| Parameter name | Value |
| --- | --- |
| n_classes | 2 |
| n_channels_in | 64 |
| n_channels_res | 16 |
| n_channels_skip | 32 |
| n_hidden | 32 |
| dilations | [2, 4, 8, 2, 4, 8, 2, 4, 8, 2, 4, 8] |
| kernel_size | 3 |
| bias | true |

[3]GNU AGPL v3 License

Table 4: Parameters used for the Aloha simulations (SNN).

| Parameter name | Value |
|---|---|
| n_classes | 2 |
| n_channels_in | 64 |
| n_channels_res | 16 |
| n_channels_skip | 32 |
| n_hidden | 32 |
| dilations | [2, 4, 8, 2, 4, 8, 2, 4, 8, 2, 4, 8] |
| threshold | 1.0 |
| learning_window | 0.3 |
| kernel_size | 2 |
| bias | true |
| $\tau_v$ | 2 |
| $\tau_s$ | 2 |
| weight_scaling (init) | 0.5 |

Table 5: Parameters used for the HeySnips simulations (ANN).

| Parameter name | Value |
|---|---|
| n_classes | 2 |
| n_channels_in | 64 |
| n_channels_res | 16 |
| n_channels_skip | 32 |
| n_hidden | 32 |
| dilations | [1, 2, 4, 8, 1, 2, 4, 8, 1, 2, 4, 8, 1, 2, 4, 8, 1, 2, 4, 8, 1, 2, 4, 8] |
| kernel_size | 3 |
| bias | true |

Table 6: Parameters used for the HeySnips simulations (SNN).

| Parameter name | Value |
|---|---|
| n_classes | 2 |
| n_channels_in | 64 |
| n_channels_res | 16 |
| n_channels_skip | 32 |
| n_hidden | 32 |
| dilations | [2, 4, 8, 16, 2, 4, 8, 16] |
| threshold | 1.0 |
| learning_window | 0.3 |
| kernel_size | 2 |
| bias | true |
| $\tau_v$ | 2 |
| $\tau_s$ | 2 |
| weight_scaling (init) | 0.5 |

Table 7: Parameters used for the SpeechCommands simulations (ANN).

| Parameter name | Value |
|---|---|
| n_classes | 35 |
| n_channels_in | 64 |
| n_channels_res | 16 |
| n_channels_skip | 32 |
| n_hidden | 32 |
| dilations | [1, 2, 4, 8, 1, 2, 4, 8, 1, 2, 4, 8, 1, 2, 4, 8, 1, 2, 4, 8, 1, 2, 4, 8] |
| kernel_size | 3 |
| bias | true |

Table 8: Parameters used for the SpeechCommands simulations (SNN).

| Parameter name | Value |
|---|---|
| n_classes | 35 |
| n_channels_in | 64 |
| n_channels_res | 32 |
| n_channels_skip | 64 |
| n_hidden | 128 |
| dilations | [2, 4, 8, 16, 2, 4, 8, 16, 2, 4, 8, 16] |
| threshold | 1.0 |
| learning_window | 0.3 |
| kernel_size | 2 |
| bias | true |
| $\tau_v$ | 2 |
| $\tau_s$ | 2 |
| weight_scaling (init) | 0.5 |

