# OpenReview forum: "WaveSense: Efficient Temporal Convolutions with Spiking Neural Networks for Keyword Spotting"
_ICLR.cc/2022/Conference — ICLR 2022 Submitted_

### Official Review · Reviewer_kAcR · 2021-10-24

**Correctness:** 2
**Technical Novelty And Significance:** 2
**Empirical Novelty And Significance:** 2
**Recommendation:** 3
**Confidence:** 4

**Main Review:**

This paper is hard to follow due to its tinpot writing and organization.

For example, there should be some parentheses to surround the cite or reference.

Besides, it’s necessary to give a formal introduction for the notations in the spiking neural model, such as $\tau_s$. It’s not reasonable to put the formulation about the spiking neural model to appendix. There are two large blanks wasteful in Page 6.

Fig. 2 does not corresponds to the context in Subsection 2.2.

In addition, using the WaveNet architecture, the authors should list all the trainable parameters and introduce their roles played in this model.

The points above prevent the understanding of this work.

**Summary Of The Paper:**

This paper presents the WaveSense model consists of spiking neurons and the WaveNet architecture for ultra-low power local signal processing. The authors conducted experiments on several real-world data sets for evaluating the efficacy of the proposed models.

**Summary Of The Review:**

This paper needs to strictly rewritten and organized for proofreading, and thus, I will take the score of “rejection”.

---

### Official Review · Reviewer_u5kv · 2021-11-01

**Correctness:** 2
**Technical Novelty And Significance:** 3
**Empirical Novelty And Significance:** 2
**Recommendation:** 3
**Confidence:** 4

**Main Review:**

Pros

1. Simulating temporal convolution in SNN by varying synaptic time constant is a neat idea. As already pointed out in the paper, it's expensive to use spike delays for neuromorphic computations. The concept of using different time constants to approximate the dynamics of the delays can potentially be more efficient to compute on the neuromorphic processor.

2. The motivation of using peak loss function for fast detection is interesting and may be useful for many other streaming applications which require instantaneous detection. It will be interesting to see how the improvement in detection speed using the peak loss function compares against conventional loss functions that use the last step output.

3. The proposed SNN outperformed other SNN methods on keyword spotting. This shows that SNN using temporal convolutions is more suitable for this task than other SNNs that use recurrent or fully-connected layers.


Cons

1. The paper lacks evidence to back the primary motivation of the proposed approach. The work is motivated by the need for low-power keyword spotting systems. However, the authors do not show any energy/computation cost comparison between the proposed SNN method and existing approaches. Without this comparison, there's no quantifiable advantage for the proposed SNN which invalidates the method's value. Energy efficiency is not a default property of SNN, and an SNN needs to be deployed appropriately on a neuromorphic processor to realize this advantage. The deployment is not trivial, and the efficiency highly relies on the dynamics and resource cost of the network. The authors need to either estimate the computation cost of the SNN by counting the number of operations, or deploy the SNN on a neuromorphic processor for energy benchmarking.

2. In addition, other low-power variants of deep networks, like binary neural networks, exist for keyword spotting (for example, [Liu, Bo, et al. 2019], [Peter, David, et al. 2020]). The authors need to discuss the advantages of using SNN over these approaches and compare the SNN with these low-power approaches on energy cost and accuracy.

3. The paper lacks ablation studies to show that the network's high performance comes from the proposed spiking temporal convolution mechanism. Since spiking neurons have an inherent information integration in the temporal dimension, the high accuracy is not enough to prove the temporal convolution is correctly functioning. When compared with the other SNN approaches, the result for the proposed method might be better due to the more complex network structure. Thus, the authors need to provide additional ablation studies on SNN without temporal convolution to prove the proposed mechanism works. Moreover, the author can also compare the synaptic time constant with the delay approach to show how much the approximation influences representation capacity.

4. The structure and writing of the paper need considerable improvements to reach the standards of the conference. The current structure of the introduction might be hard to follow for the readers who are unfamiliar with the work. The reviewer recommends the authors to divide the section into an introduction section and a related works section. The method section is also hard to follow with network design, implementation details, and experiment data mixed together. The description of the spiking temporal convolution lacks important details for readers to understand the method. The method section should be clear enough so that the readers can reproduce the method using just the write-up.

5. Figures in the paper lack sufficient details for understanding the proposed method better. Figure 1A is very similar to Figure 3 in [Oord, Aaron van den, et al. 2016], with no additional information on how SNN works with this network design. With the figure and descriptions in the text, it's very difficult to understand how the SNN computation is unrolled, and whether or not spiking neurons in each layer are updated simultaneously. Figure 2 is similar to Figure 4 in [Oord, Aaron van den, et al. 2016]. Although specific layers in [Oord, Aaron van den, et al. 2016] have been replaced by spiking layers, the paper didn't argue why these layers are needed in the block.

Oord, Aaron van den, et al. "Wavenet: A generative model for raw audio." 2016.

Coucke, Alice, et al. "Efficient keyword spotting using dilated convolutions and gating." 2018.

Liu, Bo, et al. "An ultra-low power always-on keyword spotting accelerator using quantized convolutional neural network and voltage-domain analog switching network-based approximate computing." 2019.

Peter, David, et al. "Resource-efficient DNNs for Keyword Spotting using Neural Architecture Search and Quantization." 2020.



**Summary Of The Paper:**

This paper proposes a Spiking Neural Network (SNN) with temporal convolutions for low-power keyword spotting. The proposed SNN architecture comes from the WaveNet ([Oord, Aaron van den, et al. 2016]) with modifications to adapt to the SNN dynamics. The idea for using such an architecture for keyword spotting originates from [Coucke, Alice, et al. 2018]. By training the SNN using Backprop Through Time (BPTT), the proposed method achieved better performance than other SNN-based methods and reached near SoA performance for the keyword spotting datasets.

Oord, Aaron van den, et al. "Wavenet: A generative model for raw audio." 2016.

Coucke, Alice, et al. "Efficient keyword spotting using dilated convolutions and gating." 2018.


**Summary Of The Review:**

Overall, the reviewer recommends not accepting the paper. The authors gave a neat idea on how to implement temporal convolution with spiking neurons efficiently. However, the overall quality of the paper does not reach the standards of the conference. First, additional experiments are needed to validate the effectiveness of the proposed method and demonstrate the advantage of the method compared to existing approaches. Second, the structure and writing of the paper need considerable improvements to make it more readable. More detailed problems of the paper are listed in the cons in the main review.

---

> ### Author Response · Authors · 2021-11-22
> **Reply**
>
> Dear reviewer,
>
> We thank you for your effort and the thorough review. We find your comments and suggestions very helpful and will consider them in the next iteration of our manuscript.

---

### Official Review · Reviewer_4EoH · 2021-11-02

**Correctness:** 2
**Technical Novelty And Significance:** 2
**Empirical Novelty And Significance:** 1
**Recommendation:** 3
**Confidence:** 4

**Main Review:**

Strengths: The results are interesting and promising, the benchmark is quite weak though.

Weaknesses:
1. The translation of WaveNet into SNN-based network itself appears correct. However, I indicate that the SNN considered is absolutely not an SNN given the use of multiple spikes per timestep. Fundamentally, spiking neurons in an SNN should communicate via one-bit/timestep data given the event-based data processing in nature. This is the common constraint on spike data among all SNN models. Otherwise, the network is equivalent to DNN unfolded in the time-domain with integer activations. In this regard, strictly speaking, the network considered in the present work is not SNN. Therefore, the comparison of this work with other SNNs is not fair at all. For a correct comparison, the authors should limit the number of spikes per timestep for a neuron should be limited to one and re-evaluate the performance.

2. This work hardly includes important fundamental aspects of the proposed network. The authors’ translation of WaveNet into SNN-based network is understandable; yet, it is not clear where the breakthrough lies, and how it works in compared with DNN-based WaveNet. Also, ablation study is missing.

3. Benchmark is weak. The authors used only relatively new datasets and compared their results with only few of previous works. There should be a number of previous works on google speech commands that should be compared.

**Summary Of The Paper:**

The authors propose a way to translate WaveNet into SNN-based network referred to as WaveSense. The performance of spiking wavenet on three different datasets was compared with a few previous results. Several technical aspects of the proposed method were reported including data preprocessing and translation into spiking neurons and so forth.

**Summary Of The Review:**

Based on my concerns listed above, I recommend for reject.

---

> ### Author Response · Authors · 2021-11-22
> **Reply**
>
> Dear reviewer,
>
> We thank you for your effort and the thorough review. We find your comments and suggestions very helpful and will consider them in the next iteration of our manuscript.

---

### Official Review · Reviewer_nmri · 2021-11-02

**Correctness:** 3
**Technical Novelty And Significance:** 2
**Empirical Novelty And Significance:** 3
**Recommendation:** 5
**Confidence:** 4

**Main Review:**

Strengths:
1. The paper demonstrates that the time constant of the leaky-integrate-and-fire (LIF) can be effectively used to implement time delays, as required for temporal convolution. This can reduce memory requirements of temporal spiking algorithms in general, compared to buffering spikes or using synaptic transmission delays.
2. Simple and standard neuron and synapse models allow the proposed SNN to be compatible with most neuromorphic processors. Therefore, it can have applications in low power AI systems.
3. The results presented show that the proposed SNN performs competitively against the other existing SNN models in keyword spotting tasks and is only slightly less accurate than the ANN models.

Weaknesses:
1. The paper provides limited technical novelty. The network architecture, peak detection, training loss, and learning algorithm are not new. The ANN architecture was used for keyword spotting by Coucke et al. 2018.  Also, the paper lacks the justification behind selecting the network architecture, training scheme etc. with quantitative or qualitative reasoning.
2. The paper proposes that the time constants be hand tuned based on the task. This requires at least these two comparisons to justify, which are missing from the paper: 1) a graph showing the effect of varying the time constant on accuracy for different tasks, i.e. to show quantitatively the relation between tasks and time constants and 2) comparison of the hand tuned time constant versus learning the time constant, similar to SRNN by Yin et al. 2020.
3. The paper does not quantify the improvement in memory usage from the alternative design, i.e. temporal convolution using spike buffers or using synaptic delays. The significance of the memory saved via the proposed method compared to the memory [available in standard neuromorphic processors is not known.
4. Since keyword recognition is based on maximum neuron activation during presentation, it is not clear in deployment how this proposed method will handle the absence of any keywords, because there will always be a maximum activation and intuitively a thresholding is required to avoid any false positives.
5. Since the prediction by the proposed SNN is a factor of the network state, its prediction performance is dependent on the order in which the test set is presented. The  paper should present accuracy results as the average accuracy over a large number of runs, where in each run, the test set is presented in a new random order.

**Summary Of The Paper:**

The paper proposes a spiking neural network (SNN) that takes advantage of the time constant of the neuron spiking dynamics to implement temporal convolution, instead of using synaptic delays, in order to reduce memory requirements. Input temporal data, such as audio streams, are processed continuously without any buffering, which makes it suitable for always on audio recognition systems. Further, the spiking architecture makes it suitable for low power applications using neuromorphic processors. It adopts an existing artificial neural network (ANN) architecture for generating audio waveforms, converts it to the spiking domain, and trains the network using an established spiking backpropagation algorithm. The experimental results show improved audio keyword recognition performance compared to the existing spike based methods.

**Summary Of The Review:**

While the presented results show that the proposed approach of temporal dilated spiking convolutional network can be trained to recognize audio keywords with competitive accuracy, the main contribution of the paper is a system that combines ideas from multiple existing works. Some key experiments pertaining to the claims are missing, such as the benefit of manual time constant selection or how the manual approach generalizes to new tasks, how it detects the no keyword case in deployment etc. The experiment section is missing data about memory requirement reduction than alternative approaches and throughput comparisons on a standard neuromorphic processor.

The paper can be strengthened by the addition of the following:
1) Time constant optimization instead of hand selection and its effect on accuracy on different tasks. In the current setup, it should analyze the  effect of varying the constant in Equation 8.

2)  Compare the memory requirement of the same WaveSense equivalent implemented using synaptic delays vs. WaveSense using time constants, to show the reduction in memory and computation requirements using synaptic time constant as mentioned in Section 2.1.

3) Experimental results to show robustness regarding Weakness 5.

If detailed experiments and analysis are presented on the above issues in the revised version, with justification of the related claims made, I will be inclined to reconsider my score to a marginal accept.

---

> ### Author Response · Authors · 2021-11-22
> **Reply**
>
> Dear reviewer,
>
> We thank you for your effort and the thorough review. We find your comments and suggestions very helpful and will consider them in the next iteration of our manuscript.

---

### Official Review · Reviewer_SLVL · 2021-11-03

**Correctness:** 2
**Technical Novelty And Significance:** 3
**Empirical Novelty And Significance:** 2
**Recommendation:** 3
**Confidence:** 4

**Main Review:**

Strengths:

1. This study addresses a valid problem; because although neuromorphic systems have the obvious advantage of low-power usage, regarding the performance, existing neuromorphic solutions are not at par with traditional methods.

2. Authors clearly explain how they transform the original WaveNet architecture to their SNN architecture, WaveSense, to realize an efficient neuromorphic system. I think that the experimentation configurations are also detailed sufficiently for reproducibility of the results. Authors will open source their code as well.

3. Performance evaluations are comprehensive by testing on three different public datasets, and by comparing against both existing SNN and ANN methods. The results adequately support the performance claims of the proposed model.

4. Discussion section is quite informative.


Weaknesses:

1. I think the main weakness of this paper is the limited evaluations regarding the efficiency. I understand that neuromorphic hardware is by nature low-power compared to classical digital processors, and thus we may accept that they are always efficient. However, authors do not compare the efficiency of their method with respect to previous SNN models, except the model size and parameter count in Table 1. As efficiency is the selling point, it is desirable to see the comparison against the SNN models given in Table 2 as well. Also, it would be helpful if authors can elaborate more on if neuron and parameter counts are directly comparable across different neuromorphic systems.

2. Authors often use the term “spatio-temporal”, as in “... we focus on audio tasks as spatio-temporal tasks…”. I don't quite understand the use of this term in the context of this paper because there is no spatial domain with a single channel audio. Authors should clarify this issue.

3. Authors obtain 0.95 FAPH with a 0.8% FRR, while (Coucke et al., 2018) obtains FRR 0.12% for a fixed FAPH of 0.5, on the HeySnips dataset. Authors say that their results were slightly worse. However, it is not easy to compare these numbers since authors do not fix their FAPH score. Authors need to explain how they reach that conclusion based on these numbers. A plot of FAPH versus FRR could be helpful.


**Summary Of The Paper:**

This paper proposes spiking neural networks (SNNs) as an efficient neuromorphic alternative to dilated temporal convolutions, based on the WaveNet architecture for keyword spotting. The main idea is to model the delay periods in dilated convolutions of WaveNet as synaptic time constants in SNNs for efficient implementation with neuromorphic hardware. For this task, authors introduce an effective transformation of the original WaveNet architecture to a qualitatively equivalent SNN architecture which they named as WaveSense. Then they show by experimental evaluation that WaveSense surpasses the SOTA SNN performances and come close to the SOTA performance of CNN and LSTM based methods, in keyword spotting.

**Summary Of The Review:**

Although the proposed method looks promising and comparative evaluations show performance improvements compared to SNNs nearing the ANN performances, the efficiency aspect compared to SNNs have not been justified well and there are some other critical flaws as pointed by the other reviewers. Authors did not address these issues. Therefore my decision is reject.

---

> ### Author Response · Authors · 2021-11-22
> **Reply**
>
> Dear reviewer,
>
> We thank you for your effort and the thorough review. We find your comments and suggestions very helpful and will consider them in the next iteration of our manuscript.

---

### Decision · Program_Chairs · 2022-01-20

**Decision:**

Reject

**Comment:**

The authors propose in this manuscript to use spiking neural networks (SNNs) as an efficient alternative to dilated temporal convolutions. They propose to utilize the membrane time constant of neurons instead of synaptic delays for memory efficiency. Training such networks with BPTT achieves better performance than other SNN-based methods and achieve close to SOTA compared to ANN solutions for keyword spotting.

Pros:
- The manuscript addresses an interesting problem.
- Performance is good

Cons:
- Limited evaluations regarding efficiency, although this is a main point of the paper.
- The technical novelty is limited.
- One reviewer noted that the model is not actually an SNN, due to the use of multiple spikes per time step.
- Benchmarking is weak. Little comparison with previous work.
- Structure and writing of the paper needs improvement.

The authors did not reply to any of these critical points. In summary, although the idea seems interesting, the manuscript is not ready for publication.